# Non-Invertible Defects in 5d, Boundaries and Holography

**Jeremias Aguilera Damia, Riccardo Argurio, Eduardo Garcia-Valdecasas.**

*Physique Théorique et Mathématique and International Solvay Institutes*
*Université Libre de Bruxelles; C.P. 231, 1050 Brussels, Belgium*

ABSTRACT: We show that very simple theories of abelian gauge fields with a cubic Chern-Simons term in 5d have an infinite number of non-invertible codimension two defects. They arise by dressing the symmetry operators of the broken electric 1-form symmetry with a suitable topological field theory, for any rational angle. We further discuss the same theories in the presence of a 4d boundary, and more particularly in a holographic setting. There we find that the bulk defects, when pushed to the boundary, have various different fates. Most notably, they can become codimension one non-invertible defects of a boundary theory with an ABJ anomaly.

## 1   Introduction and Summary

A modern notion of symmetry, first presented in [1], defines a symmetry by a set of topological operators and their fusion rules. Usual symmetries are generated by topological operators of codimension 1, $U_g(\Sigma_{d-1})$, obeying a group fusion law $U_{g_1}(\Sigma_{d-1}) \times U_{g_2}(\Sigma_{d-1}) = U_{g_1 \cdot g_2}(\Sigma_{d-1})$. By relaxing these two properties (codimension and group law), symmetries can be generalized in several ways. In particular, we will be interested in $p$-form symmetries $G^{(p)}$ generated by codimension $(p+1)$ operators that may obey a group law or a more general non-invertible fusion law.

Symmetries obeying a non-invertible fusion law, usually called non-invertible symmetries, have been studied in 2d QFT's [2–14]. They have also been recently realized in higher dimensions by introducing duality defects, a generalization of the usual Krammers-Wannier duality defect [15–20], and condensation defects from gauging $p$-form symmetries in submanifolds [21]. Further examples in higher dimensions have been studied in [22–26]. Interestingly, absence of non-invertible symmetries in Quantum Gravity has been shown to be necessary to argue for the completeness hypothesis [27, 28], see also [29], suggesting that the relevant notion of symmetry in Quantum Gravity is the one proposed in [1].

A particular instance of condensation/dualization defects was presented in [30, 31], where theories with an infinite number of such non-invertible defects were discussed. In those works they studied 4d theories with a global symmetry $U(1)_A^{(0)}$ suffering from an ABJ anomaly

$$d \star j_A^{(1)} = \frac{k}{8\pi^2} f^{(2)} \wedge f^{(2)} \; , \tag{1.1}$$

and showed that while $U(1)_A^{(0)}$ is broken to a $\mathbb{Z}_k^{(0)}$ subgroup, more symmetries remain. Transformations by rational angles $\alpha \in 2\pi\mathbb{Q}$ are still implemented by topological operators, once dressed by an appropriate TQFT. The price to pay is that the fusion algebra of the new operators becomes non-invertible. In the remainder of this text we will denote such non-invertible set of operators, together with their fusion algebra as $\Gamma_\mathbb{Q}$ for convenience.

In this note we present a generalization of this construction to theories in 5d with a mixed anomaly between a 1-form symmetry $U(1)^{(1)}$ and a gauged 0-form symmetry $U(1)^{(0)}$

$$d \star j^{(2)} = \frac{k}{8\pi^2} f^{(2)} \wedge f^{(2)} \ . \tag{1.2}$$

The simplest such model, which we present in Section 3.1, is a $U(1)$ gauge theory with action

$$S = \int -\frac{1}{2} f^{(2)} \wedge \star f^{(2)} + \frac{k}{24\pi^2} a^{(1)} \wedge f^{(2)} \wedge f^{(2)} \ . \tag{1.3}$$

We find that the electric symmetry $U(1)_e^{(1)}$ is not broken by the CS term to just $\mathbb{Z}_k^{(1)}$, rather there is a full $\Gamma_\mathbb{Q}^{(1)}$ remnant. The non-invertible defects for $\alpha = 2\pi/N$ are given by

$$\mathcal{D}_{1/N}(\Sigma_3) \equiv \int D\hat{c}\Big|_{\Sigma_3} \exp\left(i \oint_{\Sigma_3} \frac{2\pi}{N} \star_5 j_e^{(2)} + \frac{N}{4\pi} \hat{c}^{(1)} \wedge d\hat{c}^{(1)} - \frac{1}{2\pi} \hat{c}^{(1)} \wedge f^{(2)}\right) \ , \tag{1.4}$$

while for generic $\alpha = 2\pi p/N$ one replaces the Fractional Quantum Hall state theory with a $\mathcal{A}^{N,p}$ topological field theory [32], as in [30, 31].

An additional focus of this work is the realization of non-invertible symmetries in AdS$_5$ holography.[1] Previous works studying generalized symmetries in holographic bottom-up setups include [35–41]. A particularly nice example, which we study in Section 5, is a $U(1)_a \times U(1)_c$ gauge theory with action

$$S = \int -\frac{1}{2} f_a^{(2)} \wedge \star f_a^{(2)} - \frac{1}{2} f_c^{(2)} \wedge \star f_c^{(2)} + \frac{k}{8\pi^2} a^{(1)} \wedge f_c^{(2)} \wedge f_c^{(2)} \ . \tag{1.5}$$

In a compact space this theory has two non-invertible symmetries $\Gamma_{\mathbb{Q},a}^{(1)} \times \Gamma_{\mathbb{Q},c}^{(1)}$ arising as remnants of the explicitly broken electric 1-form symmetries. By putting this theory in $AdS_5$ we show that in bottom up holography, with appropriate boundary conditions, this model is dual to $U(1)$ gauge theory with a global symmetry $U(1)^{(0)}$ and an ABJ anomaly, one of the examples presented in [30, 31].

The rest of the paper is organised as follows. In Section 2 we review earlier constructions in 4d. In Section 3 we present a general discussion of how non-invertible 1-form symmetries arise in 5d theories with a mixed anomaly. We also consider two examples of our general construction. In Section 4, we carry out, as an appetizer to the holographic discussion, a general discussion of our models in the presence of boundaries. In Section 5 we present the holographic realization of our models, making precise connections between the symmetries in the bulk and those in the boundary.

---

[1]Note that global symmetries are believed to be absent in Quantum Gravity [33]. Hence, any top-down holographic setup lacks global symmetries in the bulk, see [34] for a thorough discussion. The models we present are understood as effective theories, where these concerns do not apply.

## 2 Non-invertible symmetries from ABJ anomalies in 4d

In this section we set the stage by reviewing the main aspects of the recent constructions [30, 31] in four dimensions. The case of study are $U(1)$ gauge theories possessing a 0-form $U(1)^{(0)}$ global symmetry broken by an ABJ anomaly. Note that simple examples of this kind are easy to find, ranging from QED-like models to effective descriptions along Coulomb branches in supersymmetric theories. On general grounds, one may regard these models as coming from a parent theory with two ordinary global symmetries $U(1)_A^{(0)}$, $U(1)_C^{(0)}$ linked through the following mixed anomaly

$$d \star j_A^{(1)} = \frac{k}{8\pi^2} F_C^{(2)} \wedge F_C^{(2)} \ , \tag{2.1}$$

with $j_A^{(1)}$ the (classically) conserved Noether current of $U(1)_A^{(0)}$ and $F_C^{(2)} \equiv dC^{(1)}$ with $C^{(1)}$ a background field for $U(1)_C^{(0)}$. The coefficient $k$ is in principle an arbitrary integer number.[2]

Assuming that the current $j_C^{(1)}$ is exactly conserved, one may gauge this symmetry and ask about the fate of the $U(1)_A^{(0)}$ global symmetry, now suffering a more severe anomaly

$$d \star j_A^{(1)} = \frac{k}{8\pi^2} f^{(2)} \wedge f^{(2)} = \frac{k}{2} \star j_m^{(2)} \wedge \star j_m^{(2)} \ . \tag{2.2}$$

For future convenience, we emphasized in the last equality that the anomalous conservation equation can be rephrased in terms of the *magnetic* symmetry current $j_m^{(2)} = (2\pi)^{-1} \star f^{(2)}$, which emerges after gauging $U(1)_c$. By explicitly constructing the topological symmetry defects, one may easily check that there is a discrete worth of global symmetry that survives the gauging procedure, $\mathbb{Z}_k^{(0)} \subset U(1)_A^{(0)}$. Since this will not be our main focus here, we can simplify our discussion further by taking $k = 1$.

Due to (2.2),[3] there is a natural obstruction to performing $U(1)_A^{(0)}$ transformations, or equivalently to define the symmetry defects realizing such transformations in subregions of spacetime. For instance, under a global rotation by a constant angle $\alpha \in [0, 2\pi)$, the effective action acquires a phase

$$\delta S = \frac{\alpha}{8\pi^2} \int_{M_4} f^{(2)} \wedge f^{(2)} \ , \tag{2.3}$$

where the integration is performed on the whole spacetime $M_4$. The above shift stands as a symptom of the complete breaking of $U(1)_A^{(0)}$ by the ABJ anomaly (2.2). However, one may try to insist on keeping (a subset of) these rotations by looking for a further operation which may compensate (2.3), such that the joint action remains a symmetry of the theory. This additional operation should be non-trivial but mapping the theory to a dual theory. These sort of operations have a longstanding history in quantum field theory and are known

---

[2]This number is usually constrained depending both on the detailed structure of the theory and the global features of spacetime. We will not delve into these particularities here, trying to keep the discussion as simple as possible.

[3]We are assuming that the spacetime topology has enough structure so as to allow for non-trivial bundles of the gauge field, as it is standard in these kind of analysis.

as duality transformations, with a simple realization in the classical electromagnetic duality. Certain duality maps can be attained by gauging a subgroup of the global symmetry of a given theory. As such, they generically modify the global structure of the theory, hence not standing as a proper symmetry, but mapping to a different (dual) theory. A simple manifestation of this fact is that, under a generic duality transformation, the effective action may produce a phase just like (2.3) (albeit for certain values of $\alpha$), which is precisely what we are looking for. The combined action of this operation and the anomalous global rotation maps the theory to itself, hence being a self-duality.

In order to make this more precise, we will explicitly implement the duality transformation appropriate for a $U(1)$ gauge theory in four dimensions. As mentioned already, a crucial role is played by the magnetic $U(1)_m^{(1)}$ 1-form symmetry. This symmetry can be coupled to background fields by turning on a 2-form $U(1)$ gauge field $B_m^{(2)}$. In particular, the duality transformation we are interested in arises by gauging a $\mathbb{Z}_N^{(1)}$ subgroup of $U(1)_m^{(1)}$. In order to do this, we single out a flat 2-form connection $b^{(2)} \in B_m^{(2)}$, with the $\mathbb{Z}_N$ condition imposed by a 1-form gauge field Lagrange multiplier $\hat{c}^{(1)}$. Moreover, this gauging is usually sensitive to certain global choices, such as the addition of discrete counterterms (sometimes referred to as discrete torsion elements). Finally, the gauged theory possesses an emergent *quantum* symmetry $\hat{\mathbb{Z}}_N^{(1)}$ which we might couple to a background field $\hat{B}^{(2)}$ (with some discrete counterterms for this field as well). This field will not play a crucial role in the following,[4] so we will set it to zero for simplicity. Putting all together, the discrete gauging is implemented by adding the following term to the effective action[5]

$$\delta S = \int_{M_4} \frac{N}{2\pi} b^{(2)} \wedge d\hat{c}^{(1)} + \frac{1}{2\pi} b^{(2)} \wedge f^{(2)} + \frac{Np'}{4\pi} b^{(2)} \wedge b^{(2)} \ . \tag{2.4}$$

In the above expression, $\hat{c}^{(1)}$ is a Lagrange multiplier setting $b^{(2)}$ to have $\mathbb{Z}_N$ periods, as it should. In addition, we introduced the integer $p'$, which is defined mod $N$, and whose role will become clear in a moment.

By "completing squares" and integrating out $\hat{c}^{(1)}$ and $b^{(2)}$, it is easy to show that,

$$\delta S = -\frac{p}{4\pi N} \int_{M_4} f^{(2)} \wedge f^{(2)} \ , \tag{2.5}$$

where $p$ is such that $pp' = 1$ mod $N$. We therefore conclude that the above phase can be used to cancel (2.3) for any given rational phase of the form $\alpha = 2\pi p/N$. The joint action of

---

[4]Ultimately, one would like to regard this operation as a symmetry transformation. A necessary condition for this to be so is for the emergent global symmetry to agree with the global symmetry of the ungauged theory. In general spacetime dimension $d$, gauging a $\mathbb{Z}_N^{(q)}$ $q$-form symmetry leads to an emergent $(d-q-2)$-form $\mathbb{Z}_N$ global symmetry [32]. In order for these to match, the condition $2q = d-2$ must be satisfied, for which $q = 1$, $d = 4$ corresponds to the example considered here. We will comment on this again when studying the five dimensional theories of our concern in this paper.

[5]See [30] for a detailed implementation of this operation in terms of modular transformations $S, T \in SL(2, \mathbb{Z})$ composed with charge conjugation.

the anomalous $U(1)_A^{(0)}$ rotation (restricted to rational phases) and the gauging (2.4) may be then regarded as a symmetry of the theory.

In order to further characterize this exotic symmetry, we need to construct the topological defects that implement it. A possible way to do it, which naturally comes out from the above procedure, is to follow the same steps but restricting ourselves to a transformation defined over half of spacetime. This will produce a duality defect implementing the symmetry. A configuration of this kind is readily obtained by defining a codimension 1 submanifold $\Sigma_3$ such that it splits $M_4$ into two disjoint sets $M_4^L$ and $M_4^R$. We then proceed as explained in this section, but only over $M_4^R$.[6] The main outcome of the gauging procedure is that now it also induces a non-trivial TQFT localized on $\Sigma_3$. Formally, this three dimensional theory is denoted by $\mathcal{A}^{N,p}$ and has the following defining property: it possesses a 1-form global symmetry $\mathbb{Z}_N^{(1)}$ with an anomaly characterized by inflow as

$$S_{anom}^{(N,p)} = -\frac{2\pi p}{2N} \int_{X_4 \, / \, . \, \partial X_4 = \Sigma_3} \mathcal{P}\left(\tilde{B}^{(2)}\right) \ , \tag{2.6}$$

with $\tilde{B}^{(2)}$ a proper $\mathbb{Z}_N$ 2-cocycle coupled to the 1-form symmetry and $\mathcal{P}$ denotes the standard Pontryagin square. Upon identifying $2\pi\tilde{B} = f \mod N$, this is precisely the anomalous shift (2.3) for $\alpha = 2\pi p/N$. In general, the topological theories $\mathcal{A}^{N,p}$ do not have a precise Lagrangian representation in three dimensions. However, for $p = 1$ ($p' = 1$), one can get such a representation of $\mathcal{A}^{N,1}$ by integrating out $b^{(2)}$ in (2.4) and keeping the Lagrange multiplier $\hat{c}^{(1)}$. In fact, neglecting the anomalous shift proportional to $f \wedge f$ one obtains

$$\int_{M_4^R} \frac{N}{2\pi} d\hat{c}^{(1)} \wedge d\hat{c}^{(1)} - \frac{1}{2\pi} d\hat{c}^{(1)} \wedge f^{(2)} = \int_{\Sigma_3} \frac{N}{2\pi} \hat{c}^{(1)} \wedge d\hat{c}^{(1)} - \frac{1}{2\pi} \hat{c}^{(1)} \wedge f^{(2)} \equiv S^{(N,1)} \ , \tag{2.7}$$

where one can recognize the gauge invariant description of the Fractional Quantum Hall State (FQHS) at filling fraction $\nu = 1/N$.

We then conclude that the joint action of an anomalous $U(1)_A^{(0)}$ rotation (with $\alpha = 2\pi/N$) and the gauging of $\mathbb{Z}_N^{(1)} \subset U(1)_m^{(1)}$ defines a symmetry implemented by the following topological defect

$$\mathcal{D}_{1/N} = \int D\hat{c}\Big|_{\Sigma_3} \exp\left(i \int_{\Sigma_3} \frac{2\pi}{N} \star_4 j_A^{(1)} + \frac{N}{2\pi} \hat{c}^{(1)} \wedge d\hat{c}^{(1)} - \frac{1}{2\pi} \hat{c}^{(1)} \wedge f^{(2)}\right) \ , \tag{2.8}$$

where we made explicit that the integration over the gauge field $\hat{c}^{(1)}$ is now restricted to $\Sigma_3$. Let us mention that the same structure can be attained by explicitly constructing a topological and gauge invariant defect consistent with the anomalous non-conservation equation (2.2). We will follow this constructive way when turning to five dimensions, subsequently making a connection with the notion of higher gauging [21].

Finally, let us briefly comment on the fusion algebra closed by elements of the form (2.8). It can be seen that they do not form a standard unitary algebra associated to a group action,

---

[6]The boundary conditions for $b^{(2)}$ on $\Sigma_3$ are Dirichlet. These, together with the flatness condition imposed by $\hat{c}^{(1)}$, are enough to show that the dependence on $\Sigma_3$ is topological, as it should [21, 30].

but instead a more general structure described by a category. We will not delve into a detailed description of this structure, but only limit ourselves to highlighting its most salient features. On the one hand, it can be checked that $\mathcal{D}_\alpha$ ($\alpha \in \mathbb{Q}$) does not have an inverse, but instead

$$\mathcal{D}_\alpha \mathcal{D}_\alpha^\dagger = \mathcal{C}_\alpha \ , \tag{2.9}$$

with $\mathcal{D}_\alpha^\dagger \equiv \mathcal{D}_{-\alpha}$ the hermitian conjugate defect of $\mathcal{D}_\alpha$, and $\mathcal{C}_\alpha$ a so-called condensation defect. Roughly speaking, $\mathcal{C}_\alpha$ can be regarded as a codimension 1 surface at which a $\mathbb{Z}_N$ subgroup of the magnetic symmetry is gauged by coupling it anti-diagonally with two dynamical gauge bundles. As such, it is transparent to local operators (for which it does close an invertible algebra) but it has a non-trivial action over magnetic lines [21]. The latter action can be deduced already at the level of $\mathcal{D}_\alpha$ and can be easily understood from the perspective of gauging in half-spacetime, as reviewed above. In fact, it is clear that, as soon as a magnetic line crosses $\Sigma_3$ to $M_4^R$, it ceases to be a gauge invariant object, due to the gauging of $\mathbb{Z}_N^{(1)} \subset U(1)_m^{(1)}$. Defining the magnetic line over a closed curve $\gamma$, this can be amended by attaching a surface $\Sigma_2$ with boundary $\gamma$. More precisely, denoting the 't Hooft line by $T(\gamma)$ then, upon crossing the symmetry defect, it gets modified as[7]

$$T(\gamma) \to \tilde{T}(\gamma, \Sigma_2) = T(\gamma) e^{-i \int_{\Sigma_2} b^{(2)}} \ . \tag{2.10}$$

Let us mention that the integral over $\Sigma_2$ can be phrased in terms of fractional magnetic flux once the connection $b^{(2)}$ is integrated out.

This concludes our brief review of how an infinite set of non-invertible symmetries stems from an anomalous conservation equation of the form (2.2) in four dimensions. In the next section, we will apply these tools to derive a similar structure in certain five dimensional gauge theories.

## 3 Non-invertible 1-form symmetries in 5d

In this section, we will argue that analogous sets of non-invertible symmetry defects can be constructed in five dimensions, though presenting some marked differences with respect to their four dimensional counterpart. The most salient one is that these defects are codimension 2, that is, they realize a non-invertible 1-form symmetry. Related to that, we will resort to a modified version of the gauging procedure described in the previous section, now restricted to a codimension 1 submanifold along the lines of [21].

Let us describe the general setup before going to the explicit description of some examples in section 3.1. In five dimensions, one may consider the following mixed anomaly between a 1-form symmetry $U(1)_B^{(1)}$ and a 0-form symmetry $U(1)_C^{(0)}$

$$d \star j_B^{(2)} = \frac{k}{8\pi^2} F_C^{(2)} \wedge F_C^{(2)} \ . \tag{3.1}$$

---

[7]We will always assume (unless explicitly said otherwise) that the curve $\gamma$ is contractible. For non-contractible $\gamma$, gauge invariance can be obtained by attaching a slightly different surface connecting the line to the defect. These subtleties will not be crucial in neither of the examples studied in this paper.

Again, assuming that $d \star j_C^{(1)} = 0$, we can gauge this symmetry and ask ourselves about the fate of the 1-form symmetry $U(1)_B^{(1)}$, whose current now satisfies

$$d \star j_B^{(2)} = \frac{k}{8\pi^2} f^{(2)} \wedge f^{(2)} = \frac{k}{2} \star j_m^{(3)} \wedge \star j_m^{(3)} \ , \tag{3.2}$$

where in the last equality we introduced the current of the magnetic symmetry $U(1)_m^{(2)}$ (note that it is a 2-form symmetry in five dimensions). As usual, one may check that there is a global $\mathbb{Z}_k^{(1)} \subset U(1)_A^{(1)}$ that is preserved by the anomaly. As we will not focus on this symmetry, we will take $k = 1$ in the following.

We will now follow a heuristic, constructive argument to show that a model featuring (3.2) actually hosts an infinite number of non-invertible codimension 2 symmetry defects. The generators of the (broken) 1-form symmetry $U(1)_B^{(1)}$ are extended operators supported on codimension 2 surfaces $\Sigma_3$ which act on line operators $L(\gamma)$, with $\gamma$ a given (closed) curve in spacetime [1]. In order for the action to be non-trivial, $\gamma$ must have non-vanishing linking with $\Sigma_3$. Let us, for the sake of concreteness, consider a fixed time slice $M_4$ and place the $U(1)_B^{(1)}$ defect

$$U_\alpha(\Sigma_3) = e^{i\alpha \int_{\Sigma_3} \star j_B^{(2)}} \ , \tag{3.3}$$

such that it splits $M_4$ into two disjoint regions. In addition, place a line operator $L(\gamma)$ such that $\gamma$ pierces $M_4$ at least once (one may think of this as the worldline of a static heavy particle). In Euclidean signature, this setup can be deformed to locally look like Figure 1a. The insertion of the line operator manifests as a delta function source along $\gamma$ for the conservation of $j_B^{(2)}$. It is then clear that the action of the symmetry defect on $L(\gamma)$ is accounted for by attaching an auxiliary surface $\Sigma_4 \subset M_4$, such that $\partial\Sigma_4 = \Sigma_3$ and integrating $d \star j_B^{(2)}$ over $\Sigma_4$. Note that the dependence on $\Sigma_4$ is topological in the sense that, as long as it is attached to $\Sigma_3$, any smooth deformation of it is equivalent. However, due to (3.2), there is an additional contribution to the naive fusion of $U_\alpha(\Sigma_3)$ and $L(\gamma)$. Formally,

$$U_\alpha(\Sigma_3)L(\gamma) = L(\gamma)e^{i\alpha}e^{i\frac{\alpha}{8\pi^2}\int_{\Sigma_4} f^{(2)} \wedge f^{(2)}} \ , \tag{3.4}$$

where we have chosen $L(\gamma)$ to have unit charge. From the above action, we see that in order to define a proper action on line operators, we need to stack $U_\alpha(\Sigma_3)$ with a TQFT. The latter must couple to the magnetic current in such a way that it cancels the unwanted term in (3.4). Again, this cannot be achieved for arbitrary phases $\alpha$, but only for the ones of the form $\alpha = 2\pi p/N$. Of course, cancelling such a term is precisely done by the $\mathcal{A}^{N,p}$ theory. We therefore get, for the particular case of $p = 1$,

$$\mathcal{D}_{1/N} \equiv \int D\hat{c}\Big|_{\Sigma_3} \exp\left(i \oint_{\Sigma_3} \frac{2\pi}{N} \star_5 j_B^{(2)} + \frac{N}{4\pi} \hat{c}^{(1)} \wedge d\hat{c}^{(1)} - \frac{1}{2\pi} \hat{c}^{(1)} \wedge f^{(2)}\right) \ . \tag{3.5}$$

Let us try to interpret this construction along the lines of section 2, namely we must look for an operation that cancels the unwanted phase in (3.4). This operation should comprise the

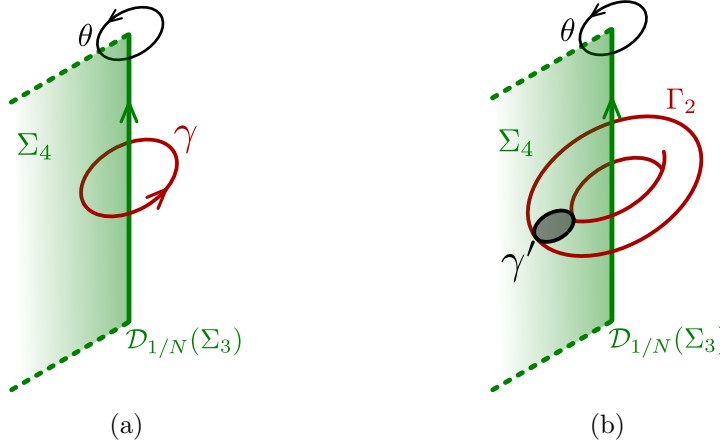

**Figure 1**: a) A non-invertible topological operator $\mathcal{D}_{1/N}$ generating a non-trivial holonomy on the linking cycle $\gamma$ has a surface $\Sigma_4$ attached to it. b) A magnetic surface $T(\Gamma_2)$ intersects $\Sigma_4$ along $\gamma'$. A flux is attached in $\Sigma_2$ such that $\partial\Sigma_2 = \gamma'$.

gauging of (a subgroup of) the magnetic symmetry. In five dimensions, such a gauging does not lead to any notion of self-duality (see footnote 4), hence one would not expect to obtain a symmetry transformation by following the exact same steps as in four dimensions. However, the fact that the anomalous phase in (3.4) has support on a codimension 1 submanifold $\Sigma_4$ hints to the fact that the appropriate gauging should be also restricted to codimension 1.

The notion of gauging a global symmetry within lower dimensional submanifolds of space-time has been recently explored in [21], where it was dubbed as *higher gauging*. A discrete $q$-form symmetry is said to be $p$-gaugeable if one can consistently sum over all possible insertions of its corresponding $(d-q-1)$-dimensional symmetry defects restricted to lie within a codimension $p$ submanifold. An obvious constraint for this to be possible is that $p \leq q + 1$. There are further constraints for the symmetry in question to be $p$-gaugeable, the so called higher anomalies, but those will not be relevant for our construction, as will become clear below.

On general grounds, $p$-gauging of a $q$-form symmetry, say $\mathbb{Z}_N$ for concreteness, amounts to summing over all non-trivial symmetry defects classified by elements in $H_{d-q-1}(\Sigma_{d-p}, \mathbb{Z}_N)$. By Poincare duality (restricted to a codimension $p$ submanifold), this is equivalent to summing over elements of $H^{q-p+1}(\Sigma_{d-p}, \mathbb{Z}_N)$, which we identify with the corresponding discrete gauge field. From this perspective, gauging a $q$-form symmetry on codimension $p$ closely resembles the gauging of a $(q-p)$-form symmetry.

We are particularly interested in gauging a $\mathbb{Z}_N^{(2)}$ subgroup of the magnetic 2-form symmetry over a codimension 1 manifold $\Sigma_4$, hence $p = 1 < q + 1$. In addition, one should check that the symmetry in question is free of higher anomalies. In the case at hand this conclusion is direct, stemming from the absence of 't Hooft anomalies for the magnetic symmetry, so

making it already 0-gaugeable[8] (a fact that holds in any spacetime dimension).

We then proceed as follows. According to the general discussion above, we need to sum over 2-form $\mathbb{Z}_N$ connections $b^{(2)}$, coupled to the magnetic symmetry along $\Sigma_4$. We then find that the problem maps to its four dimensional analog described in section 2. The appropriate operation then amounts to adding the following action supported on $\Sigma_4$

$$\delta S = \int_{\Sigma_4} \frac{N}{2\pi} b^{(2)} \wedge d\hat{c}^{(1)} + \frac{1}{2\pi} b^{(2)} \wedge f^{(2)} + \frac{Np'}{4\pi} b^{(2)} \wedge b^{(2)} \tag{3.6}$$
$$= -\frac{2\pi p}{N} \int_{\Sigma_4} f^{(2)} \wedge f^{(2)} + S^{(N,p)}[\Sigma_3] \ .$$

In the last equality we integrated out the field $b^{(2)}$ thus obtaining the corresponding term that cancels the unwanted phase in (3.4) for $\alpha = 2\pi p/N$. In addition, we formally denote by $S^{(N,p)}[\Sigma_3]$ the resulting $\mathcal{A}^{N,p}$ TQFT stacked with $U_\alpha(\Sigma_3)$ at $\Sigma_3$. For the particular case of $p = 1$, we explicitly reproduce (3.5).

Let us briefly comment on the topological nature of the defect. As emphasized above, we only require $\Sigma_4$ to satisfy $\partial \Sigma_4 = \Sigma_3$. Besides this, the actual embedding of $\Sigma_4 \subset M_5$ is totally irrelevant. One can be easily convinced about this fact by direct evaluation. Concretely, consider a four dimensional manifold $\Sigma_4'$, such that $\partial \Sigma_4' = \Sigma_3$, differing from $\Sigma_4$ by a small smooth deformation in the bulk. The difference between (3.6) evaluated in $\Sigma_4'$ and $\Sigma_4$ is then obtained by computing the partition function of the topological field theory on the closed manifold $\Sigma_4'' \equiv \Sigma_4' \cup \bar{\Sigma}_4$. The deformation being smooth implies in particular that the topology of $\Sigma_4''$ is that of the $S^4$. Finally, being a well defined topological field theory, the partition function of the theory in (3.6) is trivial on the 4-sphere.[9] In this sense, we say that the dependence on $\Sigma_4$ is topological. Moreover, the sum over $b^{(2)}$ is performed by imposing Dirichlet boundary conditions over $\Sigma_3$. This conditions, together with the fact that $b^{(2)}$ is flat, are enough to guarantee that $\Sigma_3$ can be deformed freely as a boundary of $\Sigma_4$ (see footnote 6). Putting this two facts together, one concludes that the dependence of $\Sigma_3$ is topological on $M_5$, as it should be.

Finally, we argue that non-invertible symmetry defects of this kind may act non-trivially on magnetic surfaces. The reasoning is totally analogous to the one in four dimensions. In order to make it more concrete, let us focus on the following configuration. Place a symmetry defect, say $\mathcal{D}_{1/N}$, and a magnetic surface $T(\Gamma_2)$ as shown in 1b ($\Gamma_2 \in H_2(M_5, \mathbb{Z})$). Recall that 3- and 2-manifolds do not have non-trivial linking in $d = 5$, hence the action of $\mathcal{D}_{1/N}$ over $T(\Gamma_2)$ is more subtle and lies beyond the standard definition in [1]. The surface operator intersects the attached codimension 1 surface $\Sigma_4$ along a curve $\gamma'$. Within $\Sigma_4$, a $\mathbb{Z}_N$ subgroup of the magnetic symmetry is gauged. In analogy to the four dimensional case, a consistent

---

[8]In the absence of fundamental charged matter, $U(1)$ gauge theories feature a mixed anomaly between the magnetic and the electric 1-form symmetry. However, this does not prevent gauging one of them.

[9]Equivalently, there is no non-trivial $U(1)$ instanton in the $S^4$ sourcing the second line of (3.6). Of course, these considerations do not hold if the deformation intersects a bulk magnetic surface, hence not being smooth anymore.

fusion rule demands dressing the operator by attaching a surface $\Sigma_2 \subset \Sigma_4$ such that $\partial \Sigma_2 = \gamma'$, hence the modified operator

$$T_2(\Gamma_2) \to \tilde{T}_2(\Gamma_2, \Sigma_2) \equiv T_2(\Gamma_2) e^{-i \int_{\Sigma_2} b^{(2)}} \tag{3.7}$$

is gauge invariant when intersecting $\Sigma_4$.

## 3.1 Simple examples in Maxwell-Chern-Simons theory

We now proceed to describe some five dimensional theories featuring the set of non-invertible 1-form symmetries described above. The simplest example is a $U(1)$ gauge theory with a Chern-Simons term

$$S = \int -\frac{1}{2} f^{(2)} \wedge \star f^{(2)} + \frac{k}{24\pi^2} a^{(1)} \wedge f^{(2)} \wedge f^{(2)} \ . \tag{3.8}$$

The integral is performed over an arbitrary 5-manifold $M_5$ such that there is no additional constraint on $k$ besides $k \in \mathbb{Z}$ (otherwise we should set $k \in 6\mathbb{Z}$, see for instance [42] for some discussion about this fact). Let us momentarily take $\partial M_5 = \emptyset$ in order to keep the discussion simple. The presence of a boundary has deep consequences to which we will turn our attention in section 4.

Recall that this theory possesses two higher form symmetries. On the one hand, there is a topological magnetic 2-form symmetry $U(1)_m^{(2)}$ generated by $j_m^{(3)} = (2\pi)^{-1} \star f^{(2)}$. On the other, a free $U(1)$ gauge theory has a $U(1)_e^{(1)}$ electric 1-form symmetry, with current $j_e^{(2)} = f^{(2)}$, accounting for conservation of electric charge. Turning on background fields for these symmetries, it is easy to show that they are related by a mixed anomaly accounted for by inflow as[10]

$$\frac{1}{2\pi} \int_{X_6} B_e^{(2)} \wedge dB_m^{(3)} \ . \tag{3.9}$$

For the theory (3.8), the electric symmetry is broken to $\mathbb{Z}_k^{(1)}$ by the equations of motion

$$d \star j_e^{(2)} = \frac{k}{8\pi^2} f^{(2)} \wedge f^{(2)} \ . \tag{3.10}$$

---

[10]For completeness, let us describe the anomaly corresponding to this theory, for which the electric symmetry is broken to $\mathbb{Z}_k^{(1)}$. The corresponding 2-cocycle $\tilde{B}_e^{(2)} \in H^2(M_5, \mathbb{Z}_k)$ can be embedded in $B_e^{(2)}$ as usual, $kB_e^{(2)} = 2\pi \tilde{B}_e^{(2)}$. Moreover, for general $k$, one might be interested in gauging $\mathbb{Z}_N$ subgroups of the magnetic symmetry which preserve $\mathbb{Z}_k^{(1)}$. Restricted to these discrete subgroups, the anomaly reads

$$\frac{2\pi}{k} \int_{X_6} \tilde{B}_e^{(2)} \cup \beta(\tilde{B}_m^{(3)}) \ ,$$

with $\beta : H^3(B\mathbb{Z}_N, \mathbb{Z}_N) \to H^4(B\mathbb{Z}_N, \mathbb{Z}_k)$ the Bockstein homomorphism associated to the short exact sequence $0 \to \mathbb{Z}_k \to \mathbb{Z}_{Nk} \to \mathbb{Z}_N \to 0$ and we left implicit in our notation that the inputs in the above expression are mapped to cocycles in spacetime through the usual pullback defined by the background fields. Hence the condition for the anomaly to trivialize is $\gcd(N, k) = 1$.

Equation (3.10) is precisely of the form (3.2) and arises already at the classical level in this context. As explained in the previous subsection, such a non-conservation equation leads to a set of symmetries closing on a non-invertible fusion algebra, as we will now show.

For the sake of simplicity we will set $k = 1$ in the following. Let us consider again the configuration depicted in Figure 1a. An electric 1-form symmetry transformation can be implemented by shifting the dynamical gauge field by a flat connection, locally described by

$$a^{(1)} \to a^{(1)} + d\lambda^{(0)} \quad , \quad \lambda^{(0)}(\theta + 2\pi) = \lambda^{(0)}(\theta) + \frac{2\pi p}{N} . \tag{3.11}$$

In this sense, the parameter $\lambda$ is not globally defined, but induces a non-trivial holonomy along the curve $\gamma$, locally parametrized by the angular variable $\theta$. Note that we already restrict ourselves to rational values of the induced holonomy. The actual value of $\theta$ at which the discontinuity happens determines the orientation of the auxiliary surface $\Sigma_4$. However, the topological nature of this embedding becomes manifest by realizing that it can be shifted by a trivial gauge transformation $\lambda \to \lambda + \epsilon$ with a constant $\epsilon \in [0, 2\pi)$. It is then straightforward to check that the transformation (3.11) induces the following shift in the action

$$\delta S\Big|_{\Sigma_4} = \frac{p}{4\pi N} \int_{\Sigma_4} f^{(2)} \wedge f^{(2)} , \tag{3.12}$$

hence being cancelled by the appropriate gauging of $\mathbb{Z}_N^{(2)} \subset U(1)_m^{(2)}$ within $\Sigma_4$, as explained previously.

The symmetry defects obtained this way then read

$$\mathcal{D}_{p/N} \equiv U_{\alpha = \frac{2\pi p}{N}}(\Sigma_3) \mathcal{A}^{(N,p)}(\Sigma_3) , \tag{3.13}$$

which for $p = 1$ is written more explicitly as

$$\mathcal{D}_{1/N} = \int D\hat{c}\Big|_{\Sigma_3} \exp\left( i \oint_{\Sigma_3} \frac{2\pi}{N} \star_5 f^{(2)} + \frac{N}{4\pi} \hat{c}^{(1)} \wedge d\hat{c}^{(1)} - \frac{1}{2\pi} \hat{c}^{(1)} \wedge f^{(2)} \right) . \tag{3.14}$$

Let us end our discussion on Maxwell-Chern-Simons by noting that it also has a higher group structure, similar to the one found in axion electrodynamics [43], and a related model in 3d [41]. Indeed upon coupling the theory to the backgrounds $B_e^{(2)}$ (which we take to be $\mathbb{Z}_k$-valued) and $B_m^{(3)}$, one is forced to modify the field strength $H_m^{(4)}$ of the magnetic symmetry as

$$H_m^{(4)} = dB_m^{(3)} - \frac{k}{4\pi} B_e^{(2)} \wedge B_e^{(2)} . \tag{3.15}$$

The topological defects of Maxwell-Chern-Simons hence have both non-invertible and higher group fusion rules.

Another simple example, which will be relevant when discussing holography, is the following. Consider two $U(1)$ gauge fields in five dimensions $a^{(1)}$ and $c^{(1)}$, mixed together through a Chern-Simons term

$$S = \int -\frac{1}{2} f_a^{(2)} \wedge \star f_a^{(2)} - \frac{1}{2} f_c^{(2)} \wedge \star f_c^{(2)} + \frac{k}{8\pi^2} a^{(1)} \wedge f_c^{(2)} \wedge f_c^{(2)} . \tag{3.16}$$

The equations of motion can be phrased as non-conservation equations for the electric currents $j_{e,a}^{(2)} = f_a^{(2)}$ and $j_{e,c}^{(2)} = f_c^{(2)}$

$$d \star j_{e,a}^{(2)} = \frac{k}{8\pi^2} f_c^{(2)} \wedge f_c^{(2)} \quad , \quad d \star j_{e,c}^{(2)} = \frac{k}{4\pi^2} f_a^{(2)} \wedge f_c^{(2)} \; . \tag{3.17}$$

Setting again $k = 1$, $\alpha = 2\pi/N$ for simplicity, and following the same steps as before, one can construct two sets of non-invertible operators, namely[11]

$$\mathcal{D}_{1/N}^a \equiv \int D\hat{c} \Big|_{\Sigma_3} \exp \left( i \oint_{\Sigma_3} \frac{2\pi}{N} \star_5 f_a^{(2)} + \frac{N}{4\pi} \hat{c}^{(1)} \wedge d\hat{c}^{(1)} - \frac{1}{2\pi} \hat{c}^{(1)} \wedge f_c^{(2)} \right) , \tag{3.18}$$

$$\mathcal{D}_{1/N}^c \equiv \int D\hat{c} D\hat{v} \Big|_{\Sigma_3} \exp \left( i \oint_{\Sigma_3} \frac{2\pi}{N} \star_5 f_c^{(2)} + \frac{N}{2\pi} \hat{v}^{(1)} \wedge d\hat{c}^{(1)} - \frac{1}{2\pi} \hat{c}^{(1)} \wedge f_a^{(2)} - \frac{1}{2\pi} \hat{v}^{(1)} \wedge f_c^{(2)} \right) . \tag{3.19}$$

Note that the defects described by (3.19) come equipped with two auxiliary gauge fields, $\hat{c}^{(1)}$ and $\hat{v}^{(1)}$. This is a consequence of the non-conservation equation for $j_{e,c}^{(2)}$ which involves two different magnetic currents, respectively $j_{m,a}^{(3)}$ and $j_{m,c}^{(3)}$. In fact, for general $p$ and $N$, the defects can be obtained from the following combined gauging along $\Sigma_4$

$$\int_{\Sigma_4} -\frac{Np'}{2\pi} b^{(2)} \wedge \tilde{b}^{(2)} + \frac{N}{2\pi} b^{(2)} \wedge d\hat{c}^{(1)} + \frac{N}{2\pi} \tilde{b}^{(2)} \wedge d\hat{v}^{(1)} - \frac{1}{2\pi} b^{(2)} \wedge f_c^{(2)} - \frac{1}{2\pi} \tilde{b}^{(2)} \wedge f_a^{(2)} , \tag{3.20}$$

where $p'$ is the inverse mod $N$ of $p$. Note that non-invertible defects analog to eqs. (3.18) and (3.19) exist in axion electrodynamics. In particular the codimension 1 analog to (3.18) was considered in [31]. It is also easy to show that codimension 2 operators like (3.19) exist in that theory.

## 4 Adding boundaries

In this section, we will be concerned with the dynamics of the models described above when placed on a five dimensional manifold with boundary $N_4 = \partial M_5$. Before delving into the particularities pertaining to each model, let us make a few general comments about the scope of the analysis presented below. For a general gauge theory with gauge group $G_{gauge}$ in a manifold with boundary, there are two notions of global symmetry that we want to distinguish. Firstly, there is a subgroup of the center $\Gamma \subset Z(G_{gauge})$ that generates a 1-form symmetry $\Gamma^{(1)}$, depending on the global structure of $G_{gauge}$ and on the presence of charged matter.[12] We will assume that $\Gamma^{(1)} \neq \emptyset$, as it is the case of the examples studied below. Note that $\Gamma^{(1)}$ might be either broken or preserved depending on the boundary condition imposed on the gauge bundle. Secondly, there is a special subset of the gauge transformations that act

---

[11]We hope there will be no confusion between the bulk gauge field $c^{(1)}$ and the gauge field $\hat{c}^{(1)}$ living on $\Sigma_3$ (and its extension $\Sigma_4$), which have no relation.

[12]Of course, for non simply connected gauge groups ($\pi_1(G_{gauge}) \neq 0$), there is also a $(d-3)$-form magnetic symmetry. Similar considerations apply to it, as we will mention later in this section.

non-trivially on $N_4$, denoted by $G_\partial$. These have a natural incarnation as the asymptotic gauge symmetries in holography, also referred to as *long range gauge symmetries* [34] (see Section 5).

The question about whether $G_\partial$ defines a sensible notion of global symmetry strongly depends on the choice of boundary conditions. In particular, boundary conditions that fix the gauge field at $N_4$ give $G_\partial$ a physical meaning. They are not redundancies anymore, as they act on the boundary conditions. We may regard the boundary data as a given configuration of background fields, subjected to $G_\partial$ transformations. Let us emphasize that symmetry defects associated to $G_\partial$ are genuine boundary operators and, under generic circumstances, they cannot be defined away of $N_4$. On the same note, local operators charged under $G_\partial$ are well defined only at the endpoints of bulk line operators. This is the natural perspective taken in holographic realizations, hence we will take this point of view in the following.

Finally, even if as sets, $\Gamma \subset G_\partial$, one in general does not expect a correspondence between the 1-form symmetry $\Gamma^{(1)}$ and $G_\partial$ (now regarded as a boundary global symmetry). However, as we will point out in the following sections, it is possible to establish a well defined relation between both symmetries whenever the gauge group is $G_{gauge} = U(1)$. More precisely, the action of a subset $\tilde{G}_\partial \subset G_\partial$ can be defined in terms of the action of $\Gamma^{(1)}$ over lines ending at the boundary. This map is generally not isomorphic and depends on the choice of boundary conditions. A recurrent situation that we find in presence of CS terms in the bulk is that $\Gamma^{(1)}$ becomes non-invertible. However, depending on the choice of boundary conditions, $\Gamma^{(1)}$ may still be mapped to invertible elements in $\tilde{G}_\partial$.

## 4.1 Maxwell Theory

Let us start considering 5d Maxwell theory on a manifold $M_5$ with boundary $N_4 = \partial M_5$. Once boundaries are included one needs to specify boundary conditions that fix the class of solutions one wishes to consider. A consistency condition is that variations of the fields should leave boundary conditions invariant. In the present case, a variation of the fields generates a boundary term that should vanish

$$\delta S\Big|_{N_4} = \int_{N_4} \delta a^{(1)} \wedge \star_5 f^{(2)} \ . \tag{4.1}$$

Thus, there are two choices of boundary conditions that yield a well posed variational principle

$$\text{Dirichlet: } \delta a^{(1)}\Big|_{N_4} = 0 \ , \qquad \text{Neumann: } \star_5 \, f^{(2)}\Big|_{N_4} = 0 \ . \tag{4.2}$$

Dirichlet Boundary Conditions (DBC's) fix the gauge field $a^{(1)}$ at the boundary, where it becomes a non-dynamical background. Neumann Boundary Conditions (NBC's), on the other hand, preserve gauge invariance on the whole system and $a^{(1)}$ is still a dynamical gauge field in the boundary.

Five dimensional Maxwell theory enjoys a $U(1)_e^{(1)}$ and a $U(1)_m^{(2)}$ electric and magnetic symmetries. The electric symmetry acts on Wilson Loops $W = e^{i \oint_\gamma a^{(1)}}$ and can be imple-

mented by shifting the gauge field by a flat connection $a^{(1)} \to a^{(1)} + \Lambda^{(1)}$. It is clear that Dirichlet boundary conditions, which fix $a^{(1)}|_{N_4}$, break this symmetry.

Since $U(1)_e^{(1)}$ is now broken at the boundary, we expect that imposing DBC's makes the Wilson Loops endable. Indeed, Wilson Lines are now allowed to end on the boundary while being gauge invariant, since gauge transformations[13] must leave $a^{(1)}|_{N_4}$ invariant. A Wilson Loop deep in the bulk may then move towards the boundary and open up.[14] Notice though that Wilson lines cannot open anywhere else in the bulk, so that the 1-form symmetry is not explicitly broken outside of the boundary. Instead, points at which Wilson lines end define objects naturally charged under the boundary 0-form symmetry $U(1)_\partial^{(0)}$ (see the discussion above). For the particular case of free $U(1)$ gauge theory, there is a one to one correspondence that we can establish between the symmetry defects generating $U(1)_e^{(1)}$ and $U(1)_\partial^{(0)}$. This can be achieved by noting that the action of a codimension 1 defect on a charged local operator $\mathcal{O}$ at $N_4$ can be defined in terms of the action of a codimension 2 bulk 1-form symmetry defect linked to the Wilson line ending at $\mathcal{O}$. This is consistent because, due to gauge invariance, the charge of the line under $U(1)_e^{(1)}$ and of $\mathcal{O}$ under $U(1)_\partial^{(0)}$ must agree. Moreover, both sets of operators close to isomorphic fusion algebras with $U(1)$ group law. It is in this sense that we may identify the defects of both symmetries, even if $U(1)_e^{(1)}$ is realized in the bulk while operators of $U(1)_\partial^{(0)}$ are only meaningful at $N_4$.[15] Therefore, for the free Maxwell theory on $M_5$, we have a bijective map between $\Gamma^{(1)}$ and $G_\partial$, namely $\Gamma^{(1)} \to \tilde{G}_\partial \simeq G_\partial = U(1)_\partial^{(0)}$. A more illustrative but less precise way to phrase it would be that, in this particular case, 1-form symemtry generators become identified with boundary 0-form symmetry ones when pushed to the boundary. Note that this picture does not need to hold in general, but it becomes manifestly realized in holography, where the asymptotic structure of the solutions allows to map explicitly the charges corresponding to either symmetry (see the discussion around Eq. (5.9)). For a pictorial representation of this discussion see Figure 2. As we will see next, these considerations get drastically modified in the presence of Chern-Simons terms.

In the bulk the theory also has a $U(1)_m^{(2)}$ symmetry. Magnetic surfaces charged under this symmetry cannot end on the boundary hence $U(1)_m^{(2)}$ is unbroken and it does not give rise to any sensible symmetry in $N_4$.[16]

Neumann boundary conditions are best understood in the magnetic dual picture. We introduce a dual 2-form gauge potential $\tilde{a}^{(2)}$ related to the original one as $\tilde{f}^{(3)} = \star f^{(2)}$. The $U(1)_m^{(2)}$ symmetry becomes an electric symmetry in this frame, shifting $\tilde{a}^{(2)} \to \tilde{a}^{(2)} + \tilde{\Lambda}^{(2)}$. Neumann boundary conditions in the original picture become Dirichlet in this frame, breaking

---

[13]With gauge transformations we mean gauge redundancies. Transformations acting on the boundary are physical in this case, since they modify the boundary conditions.

[14]Formally, electric line operators are classified by the relative homology group $H_1(M_5, N_4, \mathbb{Z})$ which classifies cycles closed in $M_5$ modulo its boundary.

[15]Of course, one may introduce additional matter fields restricted to the boundary, then $G_\partial$ acquires additional structure which has no correlation with any feature of $\Gamma^{(1)}$. We are not considering this situation in this paper.

[16]More precisely, one might define magnetic symmetry defects restricted to the boundary, but the DBC's precludes the presence of any non-trivial charged operator.

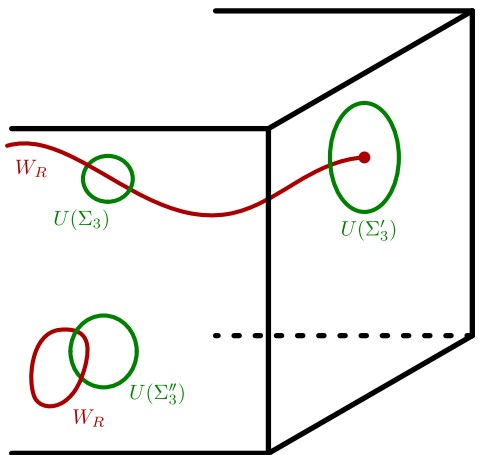

**Figure 2**: Artistic impression of 5d Maxwell Theory with boundaries. Wilson lines $W_R$ charged under $U(1)_e^{(1)}$ may end on the boundary, generating $U(1)_\partial^{(0)}$.

the $U(1)_m^{(2)}$ symmetry at the boundary and preserving $U(1)_e^{(1)}$. All the discussion above goes through interchanging the two symmetries.

## 4.2   Maxwell-Chern-Simons Theories

Let us now include a CS term and consider the theory discussed in section 3:

$$S = \int -\frac{1}{2} f^{(2)} \wedge \star f^{(2)} + \frac{k}{24\pi^2} a^{(1)} \wedge f^{(2)} \wedge f^{(2)} \ . \tag{4.3}$$

This theory is, generically, not gauge invariant in manifolds with boundaries. There are two ways to make it gauge invariant. A first approach, consists in coupling the theory to certain degrees of freedom in the boundary that cancel the gauge variation. We will not consider this possibility and will content ourselves with setting boundary conditions that guarantee gauge invariance. It is clear that this theory must be compatible only with DBC's, which restrict gauge transformations to not act on the boundary. Indeed, Neumann boundary conditions generate a dynamical gauge field in the boundary that would be anomalous by the CS term in 5d. This conclusion can also be reached by explicitly imposing the variation of the action to vanish on the boundary

$$\delta S[a^{(1)}, \delta a^{(1)}]\Big|_{N_4} = \int_{N_4} \delta a^{(1)} \wedge \star_5 f^{(2)} - \frac{k}{12\pi^2} \delta a^{(1)} \wedge a^{(1)} \wedge f^{(2)} \ . \tag{4.4}$$

Clearly the only boundary conditions making this variational problem well posed are Dirichlet. As we will mention in section 5, this obstruction is made very explicit in holography.

As detailed in section 3 the CS term already breaks $U(1)_e^{(1)}$ down to a non-invertible 1-form symmetry $\Gamma_{\mathbb{Q}}^{(1)}$ with topological operators given by eq. (3.13). Naively, by extending the arguments presented in Section 4.1 for the free theory, one is tempted to conclude that the $U(1)^{(0)}$ symmetry on the boundary suffers a similar fate. There are several ways to argue that

this is not the case and one has the standard $U(1)^{(0)}_{\partial}$ at $N_4$. Firstly, the gauge field becomes a background on the boundary, rendering the anomalous conservation eq. (3.8) harmless. There is no need to attach a TQFT to the symmetry defects located at the boundary. Furthermore, as explained in section 2, the non-invertibility of the fusion algebra arises from condensation defects having a non-trivial action on magnetic operators. With DBC's these are expelled from $N_4$ and boundary symmetry defects close to an invertible $U(1)$ algebra.

From the operator point of view, the picture is the following. In the bulk, there are symmetry defects $\mathcal{D}_{\alpha_r}$ acting with rational phases $\alpha_r \in 2\pi\mathbb{Q}$ on Wilson Loops and generating $\Gamma^{(1)}_{\mathbb{Q}}$. An obvious consequence of this fact is that there will not be a surjective map between $\Gamma^{(1)} = \Gamma^{(1)}_{\mathbb{Q}}$ and $G_{\partial} = U(1)^{(0)}_{\partial}$ (irrational phases are not accounted for in $\Gamma^{(1)}_{\mathbb{Q}}$). Accordingly, the fusion algebras do not match either. In particular, if we insist on pushing a given element of $\Gamma^{(1)}_{\mathbb{Q}}$, $\mathcal{D}_{\alpha_r}$, to the boundary, due to the absence of magnetic lines at $N_4$ the non-invertible fusion rules trivializes, and the category realizes a group law. We therefore see that the map between the 1-form symmetry and the boundary symmetry is more subtle for this theory. In particular, the non-invertible $\Gamma^{(1)} = \Gamma^{(1)}_{\mathbb{Q}}$ maps to a $\tilde{G}_{\partial}$ with the latter comprising the subset of elements of $U(1)^{(0)}_{\partial}$ accounted by rational phase rotations and, moreover, these elements have invertible fusion rules in $\tilde{G}_{\partial}$. The remaining elements of $U(1)^{(0)}_{\partial}$, namely rotations by irrational phases, are genuine boundary operators and do not necessarily map to any global structure in the bulk.

Let us consider now the model in Equation (3.16), whose action we reproduce here

$$S = \int -\frac{1}{2} f_a^{(2)} \wedge \star f_a^{(2)} - \frac{1}{2} f_c^{(2)} \wedge \star f_c^{(2)} + \frac{k}{8\pi^2} a^{(1)} \wedge f_c^{(2)} \wedge f_c^{(2)} \ . \tag{4.5}$$

Note that the CS term, even if still not invariant under $U(1)_a$ gauge transformations, is gauge invariant with respect to $U(1)_c$ in a manifold with boundary. We thus expect both DBC's and NBC's to be allowed for $c$ while $a$ should still come with DBC's and become a fixed background at $N_4$. We will denote these two boundary conditions as (D,D) and (D,N), respectively. As described in Section 3.1 this theory has $\Gamma^{(1)}_{\mathbb{Q},a} \times \Gamma^{(1)}_{\mathbb{Q},c} \times U(1)^{(2)}_{m,a} \times U(1)^{(2)}_{m,c}$ symmetry in the absence of boundaries. We now explain how these symmetries fare with the different boundary conditions. The discussion is analogous to the one for the single vector model:

- **(D,D)**: Both $\Gamma^{(1)}_a = \Gamma^{(1)}_{\mathbb{Q},a}$ and $\Gamma^{(1)}_c = \Gamma^{(1)}_{\mathbb{Q},c}$ symmetries are broken by the boundary conditions. Also like in Maxwell CS, one recovers a full $U(1)^{(0)}_a \times U(1)^{(0)}_c$ on the boundary, and the picture for each field is similar to Figure 2. The only difference is that there is a mixed anomaly between the two global symmetries on the boundary $U(1)^{(0)}_a \times U(1)^{(0)}_c$ obtained by inflow from the mixed CS term.

- **(D,N)**: In this case $\Gamma^{(1)}_a = \Gamma^{(1)}_{\mathbb{Q},a}$ is broken by the boundary condition, while $\Gamma^{(1)}_c = \Gamma^{(1)}_{\mathbb{Q},c}$ is preserved. The latter does not map to any relevant global symmetry at the boundary, since $U(1)_c$ Wilson lines cannot end at $N_4$. Regarding the magnetic symmetries, $U(1)^{(2)}_{m,a}$

and $U(1)^{(2)}_{m,c}$ are respectively preserved and broken at the boundary. Importantly, there is now a $U(1)_c$ gauge symmetry, in $N_4$. This gauge symmetry gives rise to a $U(1)^{(1)}_{m,c}$ symmetry on the boundary which is just the restriction of $U(1)^{(2)}_{m,c}$ to $N_4$, as explained for free Maxwell theory. Magnetic surfaces in the bulk may end in magnetic lines on the boundary and a similar picture to Figure 2 applies. It is due to the presence of these magnetic lines that the global 0-form symmetry $G_\partial$ becomes non-invertible. Indeed, in this case we fail to recover an $U(1)^{(0)}_{\hat{\partial}}$ at the boundary due to an ABJ anomaly, precisely as described in the introduction. Hence, we have a $\Gamma^{(1)}_a = \Gamma^{(1)}_{\mathbb{Q},a}$ symmetry in the bulk that maps bijectively to the boundary 0-form symmetry, namely $\tilde{G}_\partial \simeq G_\partial = \Gamma^{(0)}_{\mathbb{Q},a}$. Finally, we recognise the boundary theory to be just the $U(1)$ gauge theory with an ABJ anomaly introduced in [30, 31].

## 5 Implications for holography

In this section we want to investigate the relation between the non-invertible symmetries in the 5d bulk, and the symmetries with similar non-invertible structure in the boundary theory from a holographic point of view, i.e. when the bulk is $AdS_5$.

Let us begin by making a brief remark in connection with the discussion presented at the beginning of Section 4. As in a generic manifold with boundaries, in a given holographic setup, the mapping between bulk 1-form symmetry and boundary 0-form symmetry does not hold in general. In [34], the distinction between these two concepts is sharpened by the notion of *long range gauge symmetries*, associated to large gauge transformations at the boundary of $AdS$. In our context, $G_{asympt} \simeq G_\partial$. When Dirichlet boundary conditions are imposed, $G_\partial \simeq G_{gauge}$, acting on operators localized at the boundary. This is a more precise phrasing of the usual lore that boundary global 0-form symmetries are mapped to gauge symmetries in the bulk.

As mentioned in Section 4, there is no necessary connection between $G_\partial$ and the putative 1-form symemtry $\Gamma^{(1)}$ that the bulk system might enjoy. A notable exception is the $U(1)$ gauge theory in $AdS$, which is an instance in which the generators of $G_\partial$ (*i.e.* localized on the boundary) can be described by pushing the symmetry defects of the 1-form symmetry all the way to the $\partial AdS$. However, this correlation is usually broken (or at least modified) in presence of bulk Chern-Simons terms.[17]

This being said, we will proceed to discuss simple setups in which a clear connection between the 1-form symmetry and (a subset of) the boundary 0-form symmetry can be established.

---

[17]An even more decisive difference occurs when considering non-abelian gauge groups in $AdS$, say $SU(N_f)$. As usual, this is interpreted as the presence of an $SU(N_f)$ global 0-form symmetry in the dual CFT. On the contrary, for $SU(N_f)$, the bulk 1-form symmetry is just the center $\mathbb{Z}_{N_f}$, hence it is clear that operators in $\Gamma^{(1)}$ cannot coincide with the generators of $G_\partial$. This can also be understood considering the fact that $G_\partial$ can be non-abelian, while $\Gamma^{(1)}$ can only be abelian. Indeed, two codimension one defects on the boundary have a specific ordering, but as soon as one can slide them into the bulk, they become codimension two and their order can be inverted without crossing each other.

Let us consider first of all the holographic interpretation of the simplest model with one vector field. We rewrite the action for simplicity

$$S = \int -\frac{1}{2} f \wedge \star f + \frac{k}{24\pi^2} a \wedge f \wedge f \ , \tag{5.1}$$

where now it is understood that the integral is over a fixed $AdS_5$ background (we set its radius to unity)

$$ds^2 = \frac{1}{r^2} dr^2 + r^2 dx_i dx^i \ . \tag{5.2}$$

The equations of motion in components are

$$\partial_\nu \sqrt{-g} f^{\mu\nu} = -\frac{k}{32\pi^2} \epsilon^{\mu\nu\rho\sigma\alpha} f_{\nu\rho} f_{\sigma\alpha} \ . \tag{5.3}$$

The bulk field $a_\mu$ has an expansion near the boundary that is dictated by the free equation (i.e. it does not depend on $k$)

$$a_\mu(r,x) = \alpha_\mu(x) + \frac{1}{r^2} \beta_\mu(x) + \frac{1}{r^2} \log r \ \gamma_\mu(x) + \dots \tag{5.4}$$

We impose Dirichlet boundary conditions, which means that, after fixing the radial gauge $a_r = 0$, $\alpha_i$ acts as the source for a current which is given in terms of $\beta_i$. As we will see instantly, the latter statement receives corrections due to $k$. One gets the following relation from the radial equation

$$\partial_i \beta^i = -\frac{k}{16\pi^2} \epsilon^{ijkl} \partial_i \alpha_j \partial_k \alpha_l \ , \tag{5.5}$$

while the current on the boundary is identified as the coefficient of $\delta\alpha_i$ in the variation of the properly renormalized action, to be

$$j_i = -2\beta_i - \frac{k}{12\pi^2} \epsilon^{ijkl} \alpha_j \partial_k \alpha_l \ . \tag{5.6}$$

Together, these two relations yield the anomalous conservation equation

$$\partial_i j_i = \frac{k}{24\pi^2} \epsilon^{ijkl} \partial_i \alpha_j \partial_k \alpha_l \ , \tag{5.7}$$

which is the one of a global current in 4d with a cubic 't Hooft anomaly of (integer) coefficient $k$.

Now we would like to see if it is possible to switch to alternative quantization, that when $k = 0$ would amount to fixing $\beta_i$ instead of $\alpha_i$. The latter would then become a dynamical gauge field on the boundary, and $\beta_i$ its conserved current source.

When $k \neq 0$, we immediately see that there is a problem. It is not possible to add counterterms to the renormalized action in such a way to write its variation as an expression proportional to $\delta\beta_i$. This is the same conclusion as what we observed above, that it is

impossible to impose Neumann boundary conditions on the gauge field because of the cubic Chern-Simons term.

Another way to go to alternative quantization, i.e. Neumann boundary conditions, is to go to a bulk description in terms of a dual gauge field (which is then considered in ordinary quantization). In our case, we would have to describe the bulk theory in terms of a 2-form gauge field. However, it is easy to convince oneself that the cubic CS term prevents one from doing that. Therefore we conclude that the impossibility to impose Neumann boundary conditions on the vector, or to dualize it in the bulk, are holographically dual to the impossibility to gauge a symmetry with a 't Hooft anomaly.

The relation between the 1-form symmetry defects in the bulk and the 0-form ones on the boundary goes as follows. For simplicity we start with the easier case of $k = 0$. In the bulk the charge for the electric 1-form symmetry is

$$Q_e = \int_{\Sigma_3} \star J_e = \int_{\Sigma_3} \star f \ . \tag{5.8}$$

We now orient the surface $\Sigma_3$ parallel to the boundary. The components of $f$ that we should integrate over are then $f_{ri} = \partial_r a_i$, taking into account the radial gauge. We thus have

$$Q_e = \int_{\Sigma_3} d^3 x^{ijk} \sqrt{-g} \epsilon_{ijklr} \partial_r a_l = \int_{\Sigma_3} d^3 x^{ijk} r^3 \epsilon_{ijklr} \left( -2 \frac{\beta_l}{r^3} \right) = \int_{\Sigma_3} \star j \ , \tag{5.9}$$

where we have used (5.6). Note that the potentially log-divergent term actually integrates to zero on the closed surface $\Sigma_3$, since $\gamma_i \propto \partial_j f_{ij}$. We thus observe that the bulk 1-form charge exactly reduces to the charge for the 0-form symmetry on the boundary.[18]

When $k \neq 0$, the bulk defect operator is modified as described in the previous sections, and it is gauge invariant only for rational angles. When pushed to the boundary, (5.6) informs us that a similar modification is also implemented on the boundary defect operator. However at the boundary the modification is actually physically irrelevant since it is in term of background fields. The effect of the attached TQFT is to multiply the standard symmetry operator $\mathcal{U}_\theta = e^{i\theta \oint_{\Sigma_3} \star j}$ by an irrelevant phase. For the same reason, at the boundary one can define defect operators for any angle, i.e. also irrational. The latter are genuine boundary operators, i.e. they must stick to the boundary since they cannot be made gauge invariant in the bulk. We then have a proper $U(1)^{(0)}$ symmetry at the boundary.

What we learn here is the following. A 4d theory on the boundary which has a $U(1)$ symmetry with a cubic 't Hooft anomaly, possesses 3d symmetry operators that, for rational angles, can be pulled into the bulk as non-invertible defects for the electric 1-form symmetry associated to the $U(1)$ gauge bundle in the bulk (this assignation being understood in the

---

[18]One could also define a charge for the magnetic symmetry in the bulk $Q_m = \int_{\Sigma_2} \star J_m = \int_{\Sigma_2} f$. Taking again $\Sigma_2$ to the boundary and parallel to it, one ends up with $Q_m = \int_{\Sigma_2} d^2 x^{ij} \partial_i \alpha_j$, which is an expression purely in terms of background fields, and that hence acts trivially on any object in the boundary theory. We thus confirm also in this way that the bulk magnetic symmetry does not lead to any symmetry in the boundary theory.

precise sense described in Section 4). For non-rational angles, the symmetry defects cannot be pulled into the bulk, since they would cease to be gauge invariant, and hence topological. These are therefore genuine boundary operators, generating symmetries only asymptotically [34].

If we want to describe symmetry defects that are non-invertible in the boundary theory, we need to enlarge the bulk theory to the one with two vector fields.

We consider then a gauge theory in $AdS_5$ with action

$$S_5 = \int_{5d} -\frac{1}{2} f_a \wedge \star f_a - \frac{1}{2} f_c \wedge \star f_c + \frac{k}{8\pi^2} a \wedge f_c \wedge f_c \ . \tag{5.10}$$

The EOM are

$$\partial_\nu \sqrt{-g} f_a^{\mu\nu} = -\frac{k}{32\pi^2} \epsilon^{\mu\nu\rho\sigma\alpha} f_{c,\nu\rho} f_{c,\sigma\alpha} \quad , \quad \partial_\nu \sqrt{-g} f_c^{\mu\nu} = -\frac{k}{16\pi^2} \epsilon^{\mu\nu\rho\sigma\alpha} f_{a,\nu\rho} f_{c,\sigma\alpha} \ . \tag{5.11}$$

With an expansion for $c$ similar to the one for $a$, we get from the radial equations

$$\partial_i \beta_a^i = -\frac{k}{16\pi^2} \epsilon^{ijkl} \partial_i \alpha_{cj} \partial_k \alpha_{cl} \quad , \quad \partial_i \beta_c^i = -\frac{k}{8\pi^2} \epsilon^{ijkl} \partial_i \alpha_{aj} \partial_k \alpha_{cl} \ , \tag{5.12}$$

while the induced currents on the boundary are

$$j_a^i = -2\beta_a^i \quad , \quad j_c^i = -2\beta_c^i - \frac{k}{4\pi^2} \epsilon^{ijkl} \alpha_{aj} \partial_k \alpha_{cl} \ . \tag{5.13}$$

We finally get the following conservation equations

$$\partial_i j_a^i = \frac{k}{8\pi^2} \epsilon^{ijkl} \partial_i \alpha_{cj} \partial_k \alpha_{cl} \quad , \quad \partial_i j_c^i = 0 \ . \tag{5.14}$$

So we see that $j_c^i$ is conserved and generates a $U(1)_c^{(0)}$ global symmetry on the boundary, along the lines of the previous discussion. On the other hand, $j_a^i$ is not conserved in the presence of a background for $U(1)_c^{(0)}$, precisely reproducing the effect of the anomaly in the boundary theory. Hence, we may gauge $U(1)_c^{(0)}$, but this implies an ABJ anomaly for the would-be $U(1)_a^{(0)}$ symmetry in the boundary.

Gauging the $U(1)_c^{(0)}$ symmetry on the boundary amounts to imposing Neumann boundary conditions for the bulk field $c$. This is always possible, as long as one keeps imposing Dirichlet boundary conditions on $a$. Equivalently, one can dualize $c$ to a bulk 2-form potential. This has been performed recently in [40] with the aim of achieving a hydrodynamical description of the chiral anomaly. There, the breaking of the $U(1)^{(0)}$ global symmetry by the anomaly becomes manifest in the absence of $U(1)_a$ gauge symmetry for the dual system. It would be nice to look for a precise realization of the non-invertible symmetry defects in this formulation.

We note here that in the current presentation of the CS term, it is not possible to impose Neumann boundary conditions for $a$ (or to dualize it). However this state of affairs changes if one integrates by parts the CS term (i.e. adding a boundary local counterterm) so that it becomes $\propto c \wedge f_a \wedge f_c$. Now one can impose Neumann boundary conditions on $a$ but not on

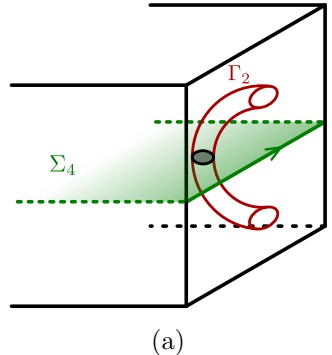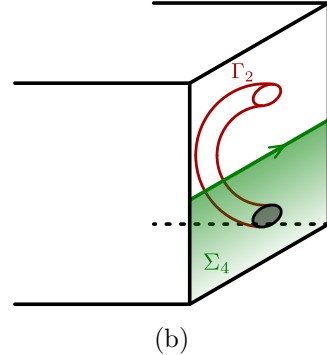

(a)                                              (b)

**Figure 3**: A non-invertible symmetry defect placed at the boundary of $AdS$. a) The auxiliary surface is extended through the bulk. b) The auxiliary surface is pushed to the boundary.

$c$. Equivalently, one can now dualize $a$. This is the holographic dual of gauging $U(1)_a^{(0)}$ in the boundary theory, which leads to a 2-group structure [39, 44].

Going back to the framework where we are gauging $U(1)_c^{(0)}$, we see that now on the boundary we are exactly in the situation where the global symmetry is broken by the ABJ anomaly. We can nevertheless build non-invertible symmetry defects in the boundary theory for any rational angle, as reviewed in section 2. It is now clear that any such defect can be pulled into the bulk, to become a non-invertible defect for the electric 1-form symmetry related to the bulk field $a$. Note that, in the bulk, there are also non-invertible defects for the electric 1-form symmetry related to $c$. Those however do not generate any global symmetry at the boundary, as explained above, due to the Neumann boundary conditions on $c$.

Let us conclude with one more comment. If we decide to build the non-invertible defects in the boundary theory by gauging a discrete subgroup of the magnetic symmetry in half of spacetime, this means that we are effectively gauging the magnetic symmetry on a four dimensional surface which ends on the defect and lies along the boundary. However as we discussed in the previous sections, the four dimensional surface can be freely moved in the bulk, as long as it ends on the defect. From this point of view, we see again that the codimension one non-invertible defects in the 4d boundary theory are just a restricted case of the codimension two defects in the 5d theory. A pictorial version of this argument is depicted in Figure 3.

### Acknowledgments

We thank Francesco Benini, Miguel Montero and especially Luigi Tizzano for helpful discussions. JAD and RA are respectively a Postdoctoral Researcher and a Research Director of the F.R.S.-FNRS (Belgium). This research is further supported by IISN-Belgium (convention 4.4503.15) and by the F.R.S.-FNRS under the "Excellence of Science" EOS be.h project n. 30820817.

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
