# Peer review of "Non-Invertible Defects in 5d, Boundaries and Holography"

_SciPost Physics_

## Round 2 · Referee Report · Anonymous (Referee 1) · 2022-10-26

Report

This is an interesting paper on a subject that is developing quickly. The goal of the paper is to study some non-invertible symmetries arising in abelian gauge theories in five dimensions with Chern-Simons terms.

The construction is a generalisation of a recent construction by Choi, Lam, Shao, and Cordova and Ohmori. It is particularly relevant in the context of holography, where such abelian theories with Chern-Simons theories often arise.

The paper is interesting, and well written, so I recommend publication after the authors address some minor points:

  1. In the paragraph below (3.6) the authors comment on the topological nature of the defect. Here it would be useful to elaborate on why the dependence on $\Sigma^4$ is only topological. At the moment this is stated with little justification, and it is not obvious to me: $p/N$ is not integral, so $p/n\int_{\Sigma^4_a - \Sigma^4_b} f^{(2)}\wedge f^{(2)}$ is not an integer for $\Sigma^4_a, \Sigma^4_b$ two manifolds with boundary $\Sigma^3$.

  2. Below (3.8) it might help the less experienced reader if it is mentioned that the fact that the Chern-Simons is well defined for $k\in\mathbb{Z}$ for any spin manifold follows from the index theorem, or alternatively some reference could be provided.

  3. In footnote 9 $\beta$ is defined as a map of cohomology classes in classifying space, but the $B$ fields are cocycles on the manifold, so the notation displayed equation in the footnote is slightly imprecise ($\beta$ does not act on $\tilde B_m^{(3)}$).

  • validity: -
  • significance: -
  • originality: -
  • clarity: -
  • formatting: -
  • grammar: -

Author:  Eduardo Garcia Valdecasas  on 2022-11-30  [id 3095]

(in reply to Report 1 on 2022-10-26)

We thank the referee for his/her careful report.

In a revised version of the paper that we will soon resubmit we have addressed the three comments of the referee.

  1. We have added the following sentence: " One can be easily convinced about this fact by direct evaluation. Concretely, consider a four dimensional manifold $\Sigma'_4$, such that $\partial\Sigma'_4=\Sigma_3$, differing from $\Sigma_4$ by a small smooth deformation in the bulk. The difference between (3.16) evaluated in $\Sigma'_4$ and $\Sigma_4$ is then obtained by computing the partition function of the topological field theory on the closed manifold $\Sigma''_4\equiv \Sigma'_4\cup \bar{\Sigma}_4$. The deformation being smooth implies in particular that the topology of $\Sigma''_4$ is that of the $S^4$. Finally, being a well defined topological field theory, the partition function of the theory in (3.16) is trivial on the 4-sphere.\footnote{\color{blue} Equivalently, there is no non-trivial $U(1)$ instanton in the $S^4$ sourcing the second line of (3.16). Of course, these considerations do not hold if the deformation intersects a bulk magnetic surface, hence not being smooth anymore.} In this sense, we say that the dependence on $\Sigma_4$ is topological." . This clarifies that, for U(1), the dependence of the operator on \Sigma_4 is topological due to the vanishing of the instanton number.

  2. We have added the following sentence below equation (3.8): "The integral is performed over an arbitrary 5-manifold M5 such that there is no additional constraint on k besides k ∈ Z (otherwise we should set k ∈ 6Z, see for instance [42] for some discussion about this fact). " Where [42] is http://arxiv.org/abs/hep-th/9603150.

  3. We have clarified the footnote, which now has a sentence: "... and we left implicit in our notation that the inputs in the above expression are mapped to cocycles in spacetime through the usual pullback defined by the background fields." This sentence clarifies the nature of the cocycles.

---

## Round 2 · Referee Report · Anonymous (Referee 2) · 2022-11-22

Report

This paper constructs non-invertible 1-form symmetries in 5d Abelian Chern-Simons theories and discusses applications in AdS/CFT. On a technical level, it provides a lift of recent constructions in 4d from a global 0-form symmetry with ABJ anomaly, building on the recently introduced notion of higher-gauging. This works adds a new and interesting chapter to a subject of wide interest and experiencing a rapid growth. The quality is good, the paper is well-written and provides appropriate references. Overall, the paper meets the requirements to be published in SciPost.

I would recommend the acceptance, provided the following points are adequately addressed. 1. From the equation in footnote 9, the map $\beta$ should take as input a 3-cocycle on $M_5$ and give as output a 4-cocyle on $X_6$, but this is not how it is defined immediately after. 2. Around Eq. (3.15), the 2-group structure is what one would expect from (3.9) after gauging the magnetic symmetry on the whole spacetime. It is not obvious to me that the 2-group structure arises also in the case. The authors may spend a few words to substantiate their claim, or to explain why the standard argument holds in the present case. 3. At the end of Section 4, in the discussion of the (D,N) case, it seems the authors discuss the non-invertible symmetry independently of the magnetic symmetry. However, being the magnetic symmetry broken by the boundary conditions, it cannot be gauged and hence the non-invertible defect cannot be constructed. I don't think it is correct to say that the non-invertible symmetry is "broken": it shouldn't be there in the first place. For the same reason, it is hard to see how the non-invertible symmetry can "map bijectively" to something on the boundary.

There are a few additional minor changes that would improve the quality and readability of the paper. 1. Abstract. - The word "cubic" on the first line can be removed: all Chern-Simons terms are cubic in 5d. - The reference to the "rational angle" is obscure if one has not read the paper. It would be useful to add a couple of words or slightly rephrase that part to make it more understandable. 2. Clarifications. - Page 3, the first time the authors mention the magnetic symmetry, they say "which emerges after gauging $U(1)_c$". It may be worth clarifying that this is the magnetic $(d-3)$-form symmetry, not the quantum symmetry, which would be a 0-form in this case. The word "emerges" may otherwise cause confusion at first. - Page 5, immediately after Eq. (2.6), it is not defined what it means that $\tilde{B}^{(2)}$ is a "proper" 2-cocyle. - In the construction of defects, in Eq.s (3.5), (3.14), (3.18) it would be useful to make it clear that the Hodge star used is the 5d one, not the one on the defect. The notation in Eq. (4.1) is most transparent. 3. Typos - Most of the times the wording "codimension" is used, but in the abstract, at the end of Section 3 (page 12), in footnote 16 (page 17), and at the end of Section 4 (page 21) , it is written "co-dimension". - Page 1, second paragraph of the introduction p-form -> $p$-form - Page 1 before Eq. (1.1), "suffering an ABJ" -> "suffering from an ABJ". - Footnote 9, "Bockstein homeomorphism" -> "Bockstein homomorphism". - Page 15 at the bottom, "there is not need" -> "there is no need" 4. Use of footnotes. - There are 17 footnotes on 20 pages, and some of them include important details to understand the arguments. I would like to suggest the authors incorporate Footnotes 4, 8, 9, 11, 17 in the main text.

Requested changes

1- Clarify $\beta$ in footnote 9. 2- Explain in more detail the compatibility of higher-group and higher-gauging. 3- Clarify the discussion of (D,N) boundary conditions at the end of Section 4.

  • validity: -
  • significance: -
  • originality: -
  • clarity: -
  • formatting: -
  • grammar: -

Author:  Eduardo Garcia Valdecasas  on 2022-11-30  [id 3096]

(in reply to Report 2 on 2022-11-22)

We are thankful to the referee for her/his in-depth reading of our work and her/his suggestions, which we have addressed as follows:

  1. We have clarified the footnote 9, which now has a sentence saying "... and we left implicit in our notation that the inputs in the above expression are mapped to cocycles in spacetime through the usual pullback defined by the background fields." This sentence clarifies the nature of the cocycles.

  2. The 2-group that we mention is unrelated to the gauging of the magnetic symmetry. We refer to a 2-group structure which is already present in the theory with action (3.8). That this theory has a 2-group is made explicit by (3.15). In this equation all fields are backgrounds. In particular, as the referee notes, we are not gauging the magnetic symmetry in the whole spacetime.

  3. We agree with the referee that the clarity of this discussion can be improved. What we meant with "broken" in the original manuscript refers to the fact that Wilson lines can open-up at the boundary. As the referee notes, this cannot happen in the bulk. We have rephrased by saying "broken at the boundary" instead of "broken". We have further added a clarifying sentence: "Notice though that Wilson lines cannot open anywhere else in the bulk, so that the 1-form symmetry is not explicitly broken outside of the boundary."

We have also implemented some of the styling suggestions of the referee.

---

## Editorial Decision

resubmitted